# Experimental Mechanical Properties and Numerical Simulation of C80 Concrete with Different Contents of Stone Powder

**DOI:** 10.3390/ma15093282

**Published:** 2022-05-03

**Authors:** Hongmei Wu, Kai Liu, Fang Yang, Bo Shen, Kejian Ma, Jiyang Zhang, Bo Liu

**Affiliations:** 1Space Structures Research Center, Guizhou University, Guiyang 550025, China; whm2019@foxmail.com (H.W.); lkyl.lbq@foxmail.com (K.L.); makejian@gzu.edu.cn (K.M.); zjy.str@foxmail.com (J.Z.); sranokada_liu@foxmail.com (B.L.); 2Key Laboratory of Structure Engineering of Guizhou Province, Guiyang 550025, China; 3College of Mining and Civil Engineering, Liupanshui Normal University, Liupanshui 553004, China; fh628909@163.com

**Keywords:** stone powder content, rock sand, inclusion theory, random aggregate distribution

## Abstract

In this paper, we show the influence of stone powder content on the mechanical properties of concrete by experiments and numerical simulations. In numerical simulation, this paper proposed a method whereby the stone powder in the numerical simulation of concrete is considered by the mechanical performances of mortar with the stone powder. The results of numerical models established based on inclusion theory and random aggregate distribution were basically consistent with the experiment, which indicated that the simulation method of concrete under different stone powder was feasible. In the range of stone powder content from 0% to 15%, the model based on inclusion theory is very close to the experimental results, and the model based on 2D random aggregate distribution is closer to the experimental value once the stone powder content is 7%. The research showed that with increased stone powder, cubic compressive strength had greater dispersion between the simulation and the experiment; axial compressive and split tensile strength reached the best levels at 5%. The best stone powder content was 5% for C80 high-strength concrete by comprehensively considering concrete’s consistency and its mechanical properties.

## 1. Introduction

With the development of the building material researches, high-performance and ultra-high-performance concrete are widely used in construction engineering, and some admixtures are added to the concrete to improve the strength [1]. Furthermore, these are also considered in order to reduce carbon dioxide emissions and economic costs. Based on this situation, many scholars have begun to replace cement with other materials to arrive at this goal [2,3]. This paper studied the effect of stone powder content on concrete strength, hoping to reduce costs by removing the manufacturing processes of washing the stone powder in rock sand.

Guizhou Province, China has a typical karst topography in which carbonate rocks are widely distributed. Due to the special geographic environment and the scarce resources of river sand, machine-made rock sand can be used as a building material by removing soil and crushing and screening rocks [4]. According to Local Code DB24/016-2010 [2], in Guizhou, the stone powder content of rock sand in concrete with strength grades C30–C45 and C50–C55 should not exceed 10% and 7%, respectively. With the development of the economy, concrete with a strength grade higher than C60 is mostly used, for which there are stringent restrictions on the content of stone powder in rock sand, although there are no clear limitations in Local Code DB24/016-2010 [5,6]. Therefore, it is important to research the limits of stone powder content in concrete with a strength grade higher than C60. This paper uses the methods of experimentation and numerical simulation to study the influence of varying stone powder contents on the mechanical properties of machine-made C80 rock sand concrete, so as to obtain the best content and provide reference data for engineering applications.

At present, the influence of stone powder content on the mechanical properties of concrete is mainly based on experimental methods. Belebchouche et al. [7], Chajec [8], and Abbasi et al. [9] researched the mechanical properties of concrete with crushed glass, granite powder, and silica stone, respectively. Demirhan et al. [10], Wang et al. [11], Diab et al. [12], and Liu et al. [13] studied the influence of different stone powder contents on the flowability and compressive strength of concrete; they found that both flowability and compressive strength decreased with increased stone powder content, but flowability was greater in concrete without stone powder. The best content was 7% for the compressive strength of C60 concrete. Chen et al. [14] studied the mechanical properties of C80 rock sand concrete and found that with increased stone powder content, the elastic modulus of the concrete had a downward trend. When the content exceeded the limit of 3% to 5%, the compressive strength of concrete below C60 declined with increased stone powder content [15,16,17,18]. The stone powder content of low-strength concrete should be 10–15%, and that of high-strength concrete should not be as high [19]. The addition of stone powder reduces the permeability of river sand concrete, but has little effect on the compressive strength [20]. Yang et al. [21] added rock chips to river sand to determine their influence on C80 concrete and found that the early age strength of high-strength rock chip concrete was higher than that of river sand concrete, whereas the long-term strength exhibited contrary results. Campos et al. [22] used stone powder and silica fume to replace Portland cement, which could decrease CO_2_ emissions, and all of the low-cement HSC they studied presented high workability. Yang et al. [23] researched the influence of different contents of basalt and limestone powders on ultra-high-performance concrete (UHPC) and found that incorporating quarry stone powder significantly improved the flowability.

The artificial intelligence and machine learning techniques are also capable of modeling the mechanical behavior of concrete. Nafees et al. [24] used machine learning techniques to build the concrete models. The nonlinearity of concrete structure calculations is undergoing extensive development. Sucharda et al. [25] indicated that the choice of parameters, the characteristics of the material, and the randomness of the model all have an important influence on nonlinear analyses. They conducted a nonlinear analysis of RC beams without shear reinforcement by studying the sensitivity of concrete material properties. Golafshani and Behnood [26] used a multi-objective ANN approach to predict the mechanical properties of concrete. Valikhani et al. [27] used constitutive relationships in a material model and determined the parameters in ATENA software, and proposed an experimental and numerical procedure to characterize the interface characteristics of concrete and their influence on the bonding strength between two materials. Karimipour et al. [28] researched the strength of concrete with different powders by experiment and numerical prediction. With regard to mesoscopic scale concrete, scholars have proposed many models, such as lattice [29,30,31], MH meso-mechanics [32,33,34], random particle [35,36,37,38], random aggregate [39,40,41,42], and random mechanical characteristic [43,44,45] models. The random aggregate model has commonly been used. Liu and Wang [46], Wang et al. [47], and Kwan et al. [48] assumed that concrete is a three-phase composite material composed of a cement mortar matrix, aggregate inclusions, and the bonding interface between the two, which defines the random aggregate model. In this model, the number of aggregates needs to be determined according to the Walraven formulation of the two-dimensional aggregate gradation curve, converted into the Fuller aggregate gradation curve [49], and then the aggregate is randomly entered into the model by the Monte Carlo method. The random aggregate model is different from other models because it can characterize the spatial random distribution of aggregate granules in concrete. Hu et al. [40] established a 2D random aggregate distribution model of expanded polystyrene concrete to study damage and failure modes with different aggregate contents. Bao et al. [41] studied the influence of the shape and content of coarse aggregate on the carbonization depth of concrete through a 2D random aggregate model. The development of concrete cracks was studied by establishing a 2D random aggregate model by Zhang et al. [42]. Wriggers and Moftah [39] established a 3D random aggregate model to study the failure behavior of concrete. Chen et al. [50] established 2D and 3D random aggregate models to simulate the basic performance of asphalt mixtures.

Most scholars have adopted experimental methods to research the mechanical properties of concrete with different stone powder contents. This paper proposes a method to simulate the influence of these mechanical properties. We mixed different contents of stone powder into mortar for testing and obtained the basic performance of the mortar to demonstrate the influence of stone powder, so as to determine the influence on the mechanical properties of concrete with different stone powder contents. These data were used in model 1, based on inclusion theory, in which concrete is regarded as a composite material of mortar and coarse aggregate wrapped in an interface, and model 2, the random aggregate model, in which concrete is considered as a composite material of mortar, coarse aggregate, and interface transition zones (ITZs). Comparing the results of model 1, model 2, and experimental values, the best stone powder content of C80 concrete was 5%. Both simulation methods can effectively simulate the impact of different stone powder contents on the mechanical properties of concrete. This paper proposes a new method to simulate the influence of stone powder content on the mechanical properties of concrete and uses the inclusion model and random aggregate distribution model to verify the feasibility of the method through comparison with experimental values.

## 2. Materials and Methods

In this section, we introduce the raw materials and experimental methods used in the experiment and analyze the results, including cubic compressive strength, axial compressive strength, elasticity modulus, split tensile strength, and flexural strength of concrete. The flowchart is shown in Figure 1.

### 2.1. Raw Materials

Conch brand P·O 42.5 cement (Conch Cement Co., Ltd., Qingzhen, China) was used in the test, and the main properties are shown in Table 1, which are tested by Chinese code GB 175-2007 [51]. The fine and coarse aggregates were limestone, both from the same area, and the additional stone powder was 200 mesh heavy calcium powder from Guizhou; the particle size distribution curves (by Mastersizer 3000—Laser Diffraction Particle Size Analyzer sourced from Spectris Instruments & Systems Shanghai Branch 1, Shanghai, China) of them are shown in Figure 2. An optimal mix ratio of C80 concrete was obtained by orthogonal tests, considering the influence of three factors and three levels: water–cement ratio (0.23, 0.25, 0.27), sand ratio (0.39, 0.41, 0.43), and mineral admixture (slag powder + fly ash + silica fume at 15% + 15% + 8%, 20% + 10% + 8%, and 25% + 5% + 8%, respectively). The water-reducing admixture is a high-performance polycarboxylate water reducer produced by Beijing Huashi Company (Beijing, China), with a water-reducing rate of more than 28%. Through the actual trial mix, the dosage of it is 1.2%, and the indicators of admixture are shown in Table 2 and Table 3. The main chemical compositions of the powders by XRF-1800 (X-Ray Fluorescence Spectrometer sourced from SHIMADZU, Shimadzu, Japan) are shown in Table 4, and the raw materials are shown in Figure 3. They were verified through experiments, and the best combinations were a water–cement ratio of 0.25, sand ratio of 0.41, and slag powder, fly ash, and silica fume in proportions of 20%, 10%, and 8% of the total cementitious material, respectively. The specific mix ratio is shown in group S0 in Table 5, where S stands for stone powder and the number represents the percentage of stone powder in the weight of rock sand; for example, S3 indicates that the stone powder content is 3% of the weight of rock sand. The mechanical parameters of cement mortar and concrete were measured by Chinese Codes JGJ/T 70-2009 (compressive strength of mortar is in its Section 9, and elastic modulus of mortar is in its Section 16) [52] and GB/T 50081-2019 (compressive strength of concrete is in Section 5, axial compressive strength of concrete is in its Section 6, and elastic modulus of concrete is in its Section 7) [53].

### 2.2. Test Method

The cubic and axial compressive, split tensile, flexural strength and elastic modulus of concrete (loading schematic and physical diagrams shown in Figure 4a,b, Figure 5a,b, Figure 6a,b and Figure 7a,b), and the axial compressive and split tensile strength and elastic modulus of mortar were tested in this study. The cubic compressive and split tensile strength tests of concrete used the same specimen size of 100 mm × 100 mm × 100 mm, the loading instrument was the same universal testing machine (3000 KN universal testing machine sourced from Ji’Nan Mts Testing Technology Co., Ltd., Jinan, China), and the loading speed was 1.0 and 0.1 MPa/s, respectively. A three-point bending experiment was conducted to test the flexural strength of concrete with a size of 100 mm × 100 mm × 400 mm, the load–deflection curves were collected by the instrument RMT-301 (Rock mechanics test system sourced from Wuhan Zhongke Kechuang Engineering Inspection Co., Ltd., Wuhan, China) automatically, and the loading speed was 0.002 mm/s. The specimen size for axial compressive strength and elastic modulus tests of concrete was 100 mm × 100 mm × 300 mm, and the loading instrument and speed were the same as in the flexural strength test. The specimen size of mortar was 70.7 mm × 70.7 mm × 220 mm, the loading instrument was the RMT-301, and the loading speed was 0.002 mm/s. The mechanical experiments were carried out on concrete and mortar with 0%, 3%, 5%, 7%, 10%, and 15% stone powder content. Before starting the experiment, we pre-loaded the equipment, rechecked the operating status of each instrument, and ensured that the instrument was running smoothly.

### 2.3. Establishment of Numerical Model

This section introduces the inclusion and random aggregate models and describes the theory of the two models, the modeling steps, and the parameter selection and settings. The 2D model was finally selected for numerical simulation by comparing the random aggregate model in 2D and 3D models under compression conditions.

#### 2.3.1. Numerical Model Based on Inclusion Theory (Model 1)

In this model, inclusion theory, homogenization of two-phase composite materials, and a progressive damage model were used to simulate the mechanical properties of concrete, as in Sun et al. [54,55]. Concrete is regarded as a composite material in which cement mortar is used as the matrix, and the coarse aggregate wrapped in the interface is used as the inclusion phase. The inclusion phase wrapped by the interface adopts the double-inclusion model shown in Figure 8. The relationship between macroscopic strain $\bar{\varepsilon }$ and stress $\bar{\sigma }$ can be transformed into that between mean strain 〈ε〉 and stress 〈σ〉 on the representative volume element (RVE), with $\bar{\varepsilon }=\langle \varepsilon \rangle$ and $\bar{\sigma }=\langle \sigma \rangle$ performed by the multi-scale method shown in Figure 9. The damage variable is represented by the damage evolution function φ(f), which can explain the relationship between the failure index f and the damage variable D in the Matzenmiller–Lubliner–Taylor (MLT) model. In this paper, the damage evolution is shown in Equations (1) and (2), where α and β are the material response parameters, and α=1, β=10.
(1)D=φ(f),0≤D≤1
(2)$$\varphi \left(f\right)=\{\begin{array}{c} 0,f<f_{\min}\\ D_{\max}\times \left(1-\exp\left(-\frac{f^{\alpha \beta }-f_{\min}{}^{\alpha \beta }}{e\beta }\right)\right) \end{array},f\geq f_{\min}$$

The concrete damage was judged by the multi-component 2D failure criteria shown in Table 6. $F_{1}$, $F_{2}$, $F_{3}$, $F_{4}$, and $F_{5}$ are the failure indices, which correspond to five failure modes. If any of the five failure indices are not less than 1, damage will occur. $X_{t}$ and $X_{c}$ are the tensile and compressive strength, respectively, of composite material in one direction, as shown in Table 4; $Y_{t}$ and $Y_{c}$ are the tensile and compressive strength, respectively, in two directions, as shown in Table 4; and S is the shear strength, which is calculated according to cubic compressive strength.

#### 2.3.2. Random Aggregate Distribution Model (Model 2)

In this study, 2D and 3D random aggregate models were used as random 2D and random 3D separately, as shown in Figure 10 and Figure 11. The white, red, and purple parts represent the mortar matrix, aggregate, and interface transition zone (ITZ), respectively. The influencing parameters include the distribution, content, and shape of the coarse aggregate, the mechanical properties of the ITZ, and the influence of porosity. Four distributions of coarse aggregate were simulated, which showed that the distribution of coarse aggregate had little effect on the mechanical properties of concrete. Therefore, the coarse aggregate was set to be uniformly distributed in the model. We simulated the area fraction of aggregate at 20%, 33%, 40%, and 50%. The value was closer to the experimental value when the area fraction was 33%. The influence of coarse aggregate shape included round, ellipse, square, and regular pentagon, and it was found that when the coarse aggregate shape was round or ellipse, the result was closer to the experiment. The mechanical property ratio of ITZ to mortar was 0.4, 0.6, 0.8, and 1, and porosity was selected as 0%, 0.5%, 1%, 1.5%, and 2%. The ratio of ITZ to mortar in the model was 0.8 and porosity was 0% after calculation and analysis. According to the gradation curve of concrete, the diameter of its coarse aggregate was determined to be 5–15 mm. Figure 12 shows that the distribution of the aggregate in the X–Y plane was random and uniform. The models of different sizes and the definitions of the relationships between aggregates were based on specific experiments. The aggregates were set to be separated from each other according to realistic conditions. The final geometric model from Digimat was exported to Abaqus for calculation.

Damage is a process of cohesion in a material that develops under loading conditions, which leads to destruction of the unit volume/interface between aggregate and mortar. The damage index was used to measure whether there is damage to concrete in the numerical model. In the two models we set up, when the damage index in the unit volume/interface arrives 0.2, the crack appears, and when it exceeds 0.9, the unit fails.

#### 2.3.3. Model Establishment

To predict the constitutive model (the stress–strain curve) of concrete based on Digimat, it was necessary to determine the mechanical parameters of the matrix phase, which were measured by experiments, as shown in Table 7. The elastic modulus and Poisson’s ratio of coarse aggregate were solved by Digimat-MX reverse regression iteration, and were set at 80 GPa and 0.16, respectively. The thickness of the ITZ was 200 μm [56], and the mechanical property ratio of ITZ to mortar was 0.8 [57,58]. The solver we used was Abaqus/Standard, and we chose static general mode. The geometric nonlinearity was off, the size of the initial increment was 1 × 10^−4^, the minimum increment size was 1 × 10^−4^, and the maximum size was 0.01. The method of the equation solver was direct, and the solution technique was full Newton. The normal contact between the steel plate and the concrete was set as hard contact, and the tangential direction was set with a coefficient of friction of 0.2. The entire loading surface of the steel plate was coupled to a reference point, and the displacement was added to it. The backing plate was restricted from rotating and moving, and the compressed steel plate could only move in the direction of the displacement.

Inclusion model (Figure 4c, Figure 5c, Figure 6c and Figure 7c): according to the failure criterion, the definition of damage variables, and the law of damage evolution, a progressive damage model of the meso characteristics of concrete materials was obtained in Digimat-MF. The calculation result of the model was imported into Abaqus as a material property of concrete in a subroutine manner, and the simulation of mechanical properties of concrete was carried out. Eight-node reduced integral solid element (C3D8R) was adopted for the concrete and steel plate in the model.

For the 2D random aggregate distribution models (Figure 4d, Figure 5d, Figure 6d and Figure 7d), the thickness of the model was 1. We set up the geometric model in Digimat-FE and then imported it into Abaqus. In order to ensure that the simulation conditions were as close as possible to the actual experimental conditions, the steel plate was established in Abaqus, and displacement was applied to it. The concrete damage plastic model was adopted for the mortar and ITZ, and the steel plate and aggregate were set to linear elasticity. For concrete with different stone powder contents in the rock sand, the same geometric model was used, and only the material properties of the mortar and ITZ were changed. All parts were simulated by plane elements, and the element type was formulated as a four-node bilinear plane stress quadrilateral element (CPS4R).

The 3D random aggregate distribution model was similar to the 2D model, but the eight-node reduced integral solid element (C3D8R) was adopted for the concrete and steel plate in the model.

As shown in Figure 13, the 2D and 3D random aggregate distribution models of concrete with 5% stone powder content were established to simulate the compressive condition. The cubic compressive strength of the concrete was 94.55 and 83.40 MPa and the axial compressive strength was 88.32 and 82.55 MPa in the 3D and 2D models, respectively. It can be seen that the cubic compressive and axial compressive strength of the 3D model exceeded the 2D model slightly with the same size coarse aggregate content, and the calculation result of the 2D model was closer to the experiment. Therefore, the 2D random aggregate model was selected to simulate the mechanical properties of concrete, as shown in Figure 4d, Figure 5d, Figure 6d and Figure 7d. The comparison of compressive strength of the two models shows that there was little difference between them, and the simulation value of 2D would be closer to the experimental value; therefore, we used the 3D random aggregate distribution model of concrete with 5% stone powder content to simulate cubic compressive strength and axial compressive strength, which verified the feasibility of the 2D model. Therefore, the 2D random aggregate distribution model was used in all tests.

## 3. Analysis of the Experiment Results

Research on the flowability of C80 concrete with stone powder content showed that flowability was the best when the content was 7%. With increased stone powder content, the slump and slump-flow decreased, and an agglomerated phenomenon occurred in the concrete. The changes in compressive, split tensile, and flexural strength of C80 concrete with different stone powder content were as follows.

### 3.1. Calculation of Experimental Results

The cubic compressive and axial compressive strength of concrete are calculated by Equation (3):(3)$$f_{ci}=F/A$$
where i=c or r, $f_{cc}$ represents the cubic compressive strength of concrete (MPa), $f_{cr}$ represents the axial compressive strength of concrete (MPa), F is the failure load (N), and A is the load area (mm^2^).

The split tensile strength of concrete is calculated by Equation (4):(4)$$f_{ts}=2F/\pi A=0.637F/A$$
where $f_{ts}$ represents split tensile strength of concrete (MPa), F is the failure load (N), and A is the load area (mm^2^).

The flexural strength of concrete is calculated by Equation (5):(5)$$f_{f}=3Fl/2bh^{2}$$
where $f_{f}$ is the flexural strength of concrete (MPa), F is the failure load (N), l shows the distance between the supports (mm), h is the section height of the specimen (mm), and b is the section width of the specimen (mm).

The mechanical parameters of cement mortar and concrete were measured by the above methods, and each group of experiments had three specimens. The test result was the average value of the three specimens according to Chinese Codes JGJ/T 70-2009 [52] and GB/T 50081-2019 [53], shown in Table 7 and Table 8. Stone powder replaced the weight of rock sand. $c_{\upsilon }$ is the coefficient of variation. The mechanical properties of the interface transition zone were 0.8 times that of the mortar, according to Zhang and Du [57] and Huang et al. [58].

### 3.2. Cubic Compressive Strength

Cubic compressive strength is an important criterion to divide the grades of concrete. From Table 8, it can be seen that as the content of stone powder increased, the strength of concrete gradually increased and reached a maximum of 97 MPa when the content was 10%, and then dropped sharply. The excessive experimental value of S10 was due to the uneven distribution of coarse aggregate [59,60]. In general, in the range of 3–10% stone powder content, the cubic strength of concrete increased with increased stone powder because, with the increase in stone powder content, the filling effect is more effective at making the concrete denser, which leads to strength increase. When the stone powder content exceeds 10%, too much stone powder content will lead to uneven distribution of aggregates and more weak parts, which causes it to decrease [61].

### 3.3. Axial Compressive Strength

The axial compressive strength test of concrete can reflect the stress and failure state intuitively. It can be seen from Table 8 that when the content of stone powder was 5%, the axial compressive strength of concrete reached the maximum value of 85.3 MPa. In the range of 5–15% content, the strength decreased gradually, because when the content of stone powder exceeds 5%, the concrete will occur agglomeration, which leads to a strength decrease [62].

### 3.4. Split Tensile Strength

The split tensile strength test is one of the methods to test the tensile strength of concrete which can reflect its tensile performance indirectly. It can be seen from Table 8 that the change trend was the same as that of axial compressive strength, reaching the maximum value of 5.1 MPa at 5% content and then decreasing gradually. It is also caused by the uneven distribution of aggregate due to excessive stone powder

### 3.5. Flexural Strength

Flexural strength can reflect the toughness of concrete. As shown in Table 8, the flexural strength was the largest, 6.69 MPa, at 3%, and then decreased, again reaching a peak of 6.46 MPa at 7%. In general, the effect of stone powder content on the flexural strength is small, as they are all within 10%, but the effect of excessive stone powder content on the reduction in strength is still the same.

### 3.6. Stress–Strain Curves

The stress–strain curves of concrete with different stone powder contents are shown in Figure 14. Failure is sudden due to the brittleness of C80 concrete, so the descending section is steep. As seen in Figure 14, the stone powder content ranges from 3 to 5%, the peak stress of concrete increases and reaches the maximum at 5%, and then, with the increased content, the peak stress shows a steady downward trend.

## 4. Comparison Analysis between Numerical Simulation and Experimental Results

This section mainly includes two parts: analysis and discussion. The analysis includes a comparative analysis of the experimental value and two simulated values and a comparison of damage and failure modes under compression in two models. The discussion includes the main research results of this paper, some abnormal data analysis, and the limitations of the paper.

### 4.1. Analysis of Results

In the following figures, M1 and M2 represent the simulation result of the numerical model based on inclusion theory and the 2D random aggregate distribution model, respectively. Figure 15 is a comparison diagram of experimental and simulation results. As seen in Figure 15a, the cubic compressive strength values between the simulation and the experiment are relatively large, because the coarse aggregate could not be distributed evenly in the experiment. Some larger or smaller size coarse aggregate affected the experimental results and caused a certain difference between the simulation and experiment. In general, the simulated value based on inclusion theory was closer to the experimental value. The stress–strain curves of concrete with different stone powder contents were similar; therefore, here we took the stress–strain curves of concrete with a content of 5% as an example for comparison. As shown in Figure 15b, the simulation results based on inclusion theory were more consistent with the experiment. From Figure 15c,d,f, the results based on inclusion theory coincide with the experiment, and their change laws are the same. Among them, axial compressive and split tensile strength reached the maximum when the content was 5%. Figure 15e shows that the simulated value of the elastic modulus was close to the experiment, but the change rule was different. The experiment showed that the elastic modulus decreased as the content increased, and both models showed a rebound when the content was 7%.

For cubic compressive strength, the experimental results are similar to the two simulation results, but the change trend is different. The test reaches the maximum result when the stone powder content is 10%, the simulation result based on the inclusion theory reaches the maximum at 5%, and the 2D random aggregate model reaches the maximum when the stone powder content is 15%. For axial compressive strength, the variation trends of the experimental results and the simulation results are the same, but the results of the model based on the 2D random aggregate distribution are smaller than those of the test when the stone powder content is 7%. For splitting tensile strength, the model based on the 2D random aggregate distribution has some difference when the stone powder content is 5%. The elastic modulus and flexural strength results of concrete are not much different between experiments and simulations.

It can be seen that the two simulation results are in good agreement with the experiment, but the model based on inclusion theory can better reflect the macro-mechanical properties of concrete, because this method is based on inclusion theory, homogenization theory, and progressive damage theory to calculate the constitutive relationship of concrete with a certain content, and then the stress–strain curves are directly imported from Digimat into Abaqus as a material parameter. Meanwhile, the model as a whole could exhibit its macroscopic mechanical properties accurately. However, the model did not consider the distribution of aggregates and mortar, and as a result, it could not reflect the mesoscopic failure mode of concrete; therefore, the establishment of a random aggregate distribution model is significant.

Figure 16 shows the cubic compression failure mode of concrete with 5% stone powder content based on inclusion theory. The final macroscopic crack is characterized by the damage index when DAMAGEC exceeds 0.9. Compression damage begins at point a of the stress–strain curve, first appears at the four corners of the concrete, as shown in Figure 16a, and then expands to point b with increasing stress, as shown in Figure 16b. The damage gradually extends to the middle of the concrete surface because the steel plates constrain the concrete due to the friction between them (Figure 16c). Full peeling occurs because, without side edges, the smaller pressure-bearing area means a reduced bearing capacity of concrete, and, finally, the concrete suffers from compression damage, as shown in Figure 16d.

Figure 17 shows the whole process of damage and failure of cubic concrete specimens in the 2D random aggregate distribution model. At point a in Figure 17, the damage first occurs at the ITZ with low stiffness and bearing capacity. The mismatched stiffness between the aggregate and interface causes a stress concentration phenomenon, and cracks appear and expand in the ITZ, as shown in Figure 17a. At pointb, there is a process of developing damage when the stress shifts from the yield stage to the maximum. The damage begins to develop to the middle of the mortar from the ITZ, and the four corners of the concrete are also damaged with increasing load (Figure 17b). At pointc, the stress is at its maximum and the model begins to fail, and the damage reaches the maximum that the interface can withstand; thus, the bearing capacity of the specimen starts to decrease. The microcracks are connected to form failure areas, as shown in Figure 17c. At point d in Figure 17d, microcrack areas develop and penetrate into macroscopic cracks with increased displacement after the peak load, and the bearing capacity drops sharply. The upper and lower parts of the specimen are not damaged much due to the constraint of the steel plates. As shown in Figure 17e, the interface is broken, the concrete cannot continue to bear pressure, and the crack has reached the maximum.

There are some differences between the two failure modes. In the model based on inclusion theory, whose first crack occurs at the four corners, the failure develops toward the middle of the specimen with increasing load. In the damage to a 2D random aggregate distribution, whose first crack appears in the ITZ, the microcracks are connected and develop into macroscopic cracks with increasing load. The final failure mode of the two models is an X shape, which is consistent with the experiment. Therefore, the nonlinear characteristics of concrete at the macro level are closely related to the initiation and propagation of microcracks in concrete.

### 4.2. Discussion

At present, there is little research on stone powder in concrete through numerical simulations. Therefore, based on previous studies, this paper proposes a simulation method for stone powder, adding different contents of stone powder to the mortar and testing its mechanical properties. The changed mechanical properties of mortar indicate the impact of the stone powder content, and the inclusion model and 2D random aggregate distribution model were established to verify the method in comparison with experimental values. The experimental and simulation results show the influence of stone powder content on the mechanical properties of concrete, and the best stone powder content of C80 concrete is 5%.

The experimental results show that with increased stone powder content, the cubic compressive strength first increases and then decreases, reaching 97 MPa when the content is 10%, which is too large compared with the other groups. This is due to the influence of concrete size. In this experiment, the size of the cube compressive specimen is 100 mm × 100 mm × 100 mm, and the sizes of the axial compressive strength and elastic modulus are both 100 mm × 100 mm × 300 mm. The smaller the specimen size, the greater the probability of uneven distribution of the coarse aggregate, which is the reason for the greater difference in the cubic compressive strengths. Comparing the experiment and the simulation, when the stone powder content was less than 7%, the values of cubic compressive strength were close, but the result of the S10 group was large. When the stone powder content exceeded 7%, the thick mortar led to uneven distribution of the coarse aggregate in the specimen during the experiment, and the coarse aggregate was concentrated in the middle. Han et al. [62] also found that the distribution of coarse aggregate has a significant effect on the strength of concrete. Huang et al. [59] also found that the different distribution of coarse aggregates has an effect on concrete carbonation. In the two models, the coarse aggregate was distributed evenly. Due to this difference, the failure mode of the specimen differed from the experimental test, which is reflected in the macroscopic strength. For axial compressive strength, when the stone powder content is before 5%, the results of the experiment and the two simulation methods are in good agreement. When the stone powder content exceeds 7%, the simulation results of the 2D random aggregate distribution model have little difference compared with the experimental values. This is because, in this model, we set the performance of the interface transition zone to be 0.8 times that of the mortar, which may be different from the actual one, causing the difference in the results. For the mechanical properties of other groups, the experiment and simulation were close.

The model established based on inclusion theory was closer to the experiment [59]. This is because, in this model, the material was damaged as a whole and was destroyed uniformly, so the overall macroscopic performance was more consistent with the experimental value. The 2D random aggregate model focused more on the distribution of internal materials and had more influencing factors during failure, which can reflect the meso-level failure mode of concrete more accurately [42].

Although this paper has proposed a method to simulate different contents of stone powder, it cannot be completely separated from the experiment, and the basic data of the experiment still need to be provided. Meanwhile, the model does not consider that the coarse aggregate in thick mortar would be distributed unevenly. Therefore, it is necessary to establish a simulation method for concrete with different stone powder contents according to the actual distribution of coarse aggregate and to analyze the influence of different distributions of coarse aggregate on the mechanical properties of concrete with different stone powder content.

## 5. Conclusions

Experimental analysis and numerical simulation methods based on inclusion theory and random aggregate distribution were used to study the influence of stone powder content on the mechanical properties of C80 concrete in this paper. The conclusions are as follows:

With increased stone powder, the cubic compressive strength of concrete showed different changes. The experimental value reached the maximum at 10%, the numerical model based on inclusion theory reached the maximum at 5%, and the 2D random aggregate distribution model reached the maximum at 15%. There was greater dispersion between experiment and simulation.

In the range of stone powder content from 0% to 15%, the model based on inclusion theory was very close to the experimental results, and the model based on 2D random aggregate distribution was closer to the experimental value after the stone powder content was 7%.

The simulated values were consistent with the experiment in the axial compressive, split tensile, and flexural strength of concrete. The model based on inclusion theory reflected the macroscopic mechanical properties of concrete, and the 2D random aggregate model emphasized the meso-level failure mode.

Considering the mechanical properties and flowability of concrete comprehensively, the best stone powder content of C80 concrete was 5%.

The final failure mode of the two models was an X shape, which was consistent with the experiment. Therefore, the nonlinear characteristics of concrete at the macro level are closely related to the initiation and propagation of microcracks in concrete. The 2D random aggregate distribution model can better show the mesoscopic mode of the compression failure of concrete.

## Figures and Tables

**Figure 1 materials-15-03282-f001:**
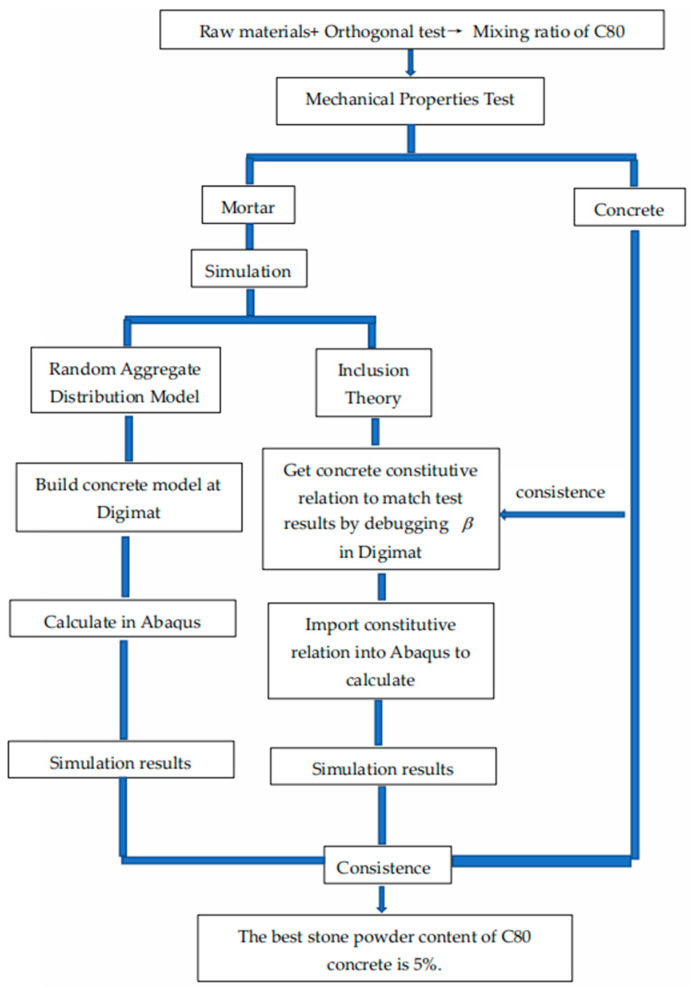
Flowchart representing the methodology of the work.

**Figure 2 materials-15-03282-f002:**
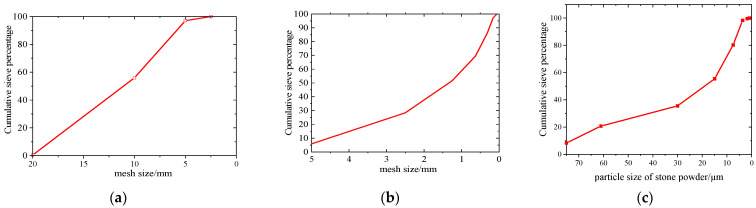
The particle size distribution curves for the aggregates: (**a**) coarse aggregate, (**b**) fine aggregate, and (**c**) stone powder.

**Figure 3 materials-15-03282-f003:**
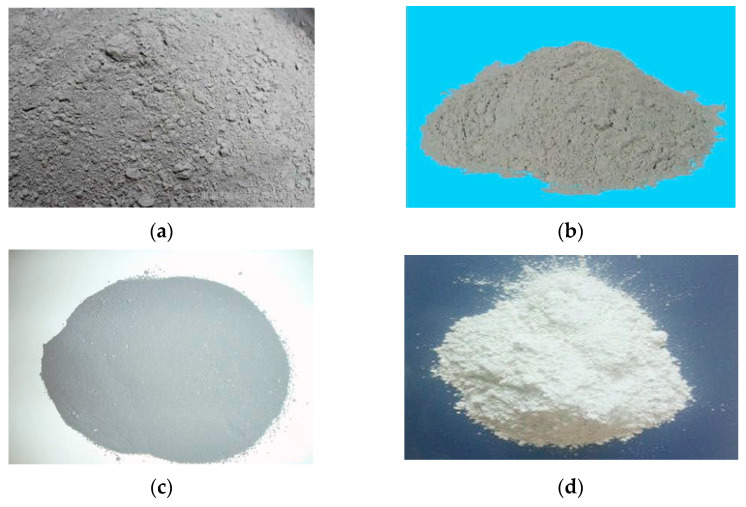
Raw materials: (**a**) fly Ash, (**b**) slag Powder, (**c**) silica Fume, and (**d**) stone Powder.

**Figure 4 materials-15-03282-f004:**
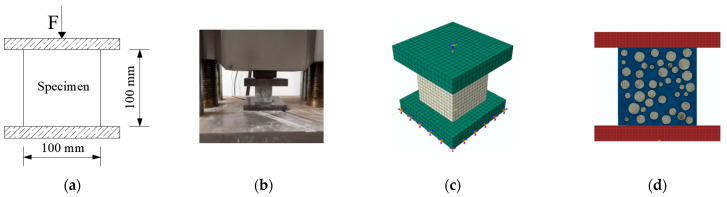
Cubic compressive strength test: (**a**) schematic design, (**b**) physical diagram, (**c**) inclusion model, and (**d**) 2D random model.

**Figure 5 materials-15-03282-f005:**
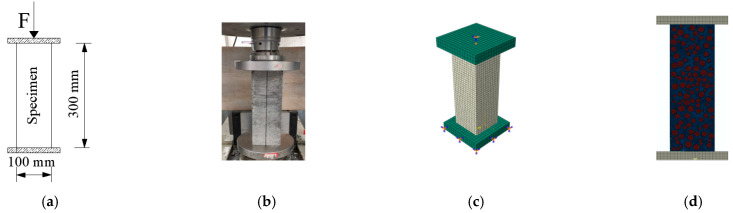
Axial compressive strength and elastic modulus test: (**a**) schematic design, (**b**) physical diagram, (**c**) inclusion model, and (**d**) 2D random model.

**Figure 6 materials-15-03282-f006:**
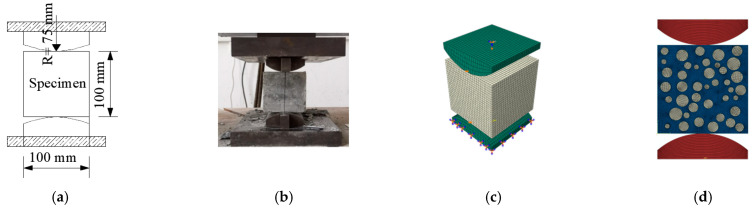
Split tensile strength test: (**a**) schematic diagram, (**b**) physical diagram, (**c**) inclusion model, and (**d**) 2D random model.

**Figure 7 materials-15-03282-f007:**
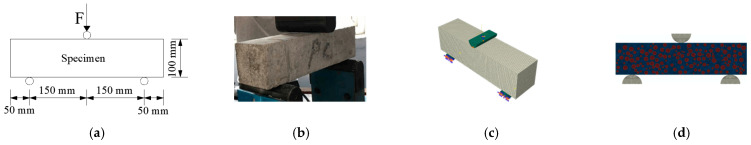
Flexural strength test: (**a**) schematic diagram, (**b**) physical diagram, (**c**) inclusion model, and (**d**) 2D random model.

**Figure 8 materials-15-03282-f008:**
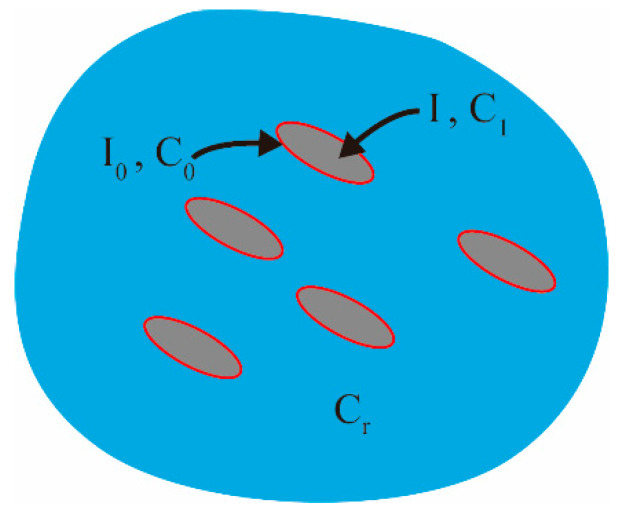
Double-inclusion model.

**Figure 9 materials-15-03282-f009:**
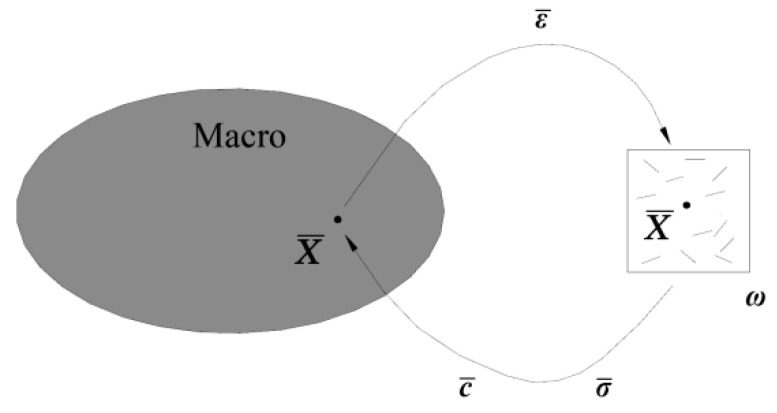
Multi-scale method.

**Figure 10 materials-15-03282-f010:**
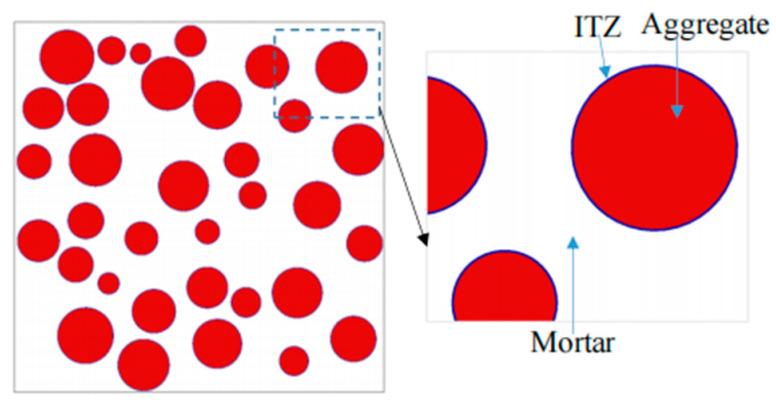
Two-dimensional random aggregate model of concrete.

**Figure 11 materials-15-03282-f011:**
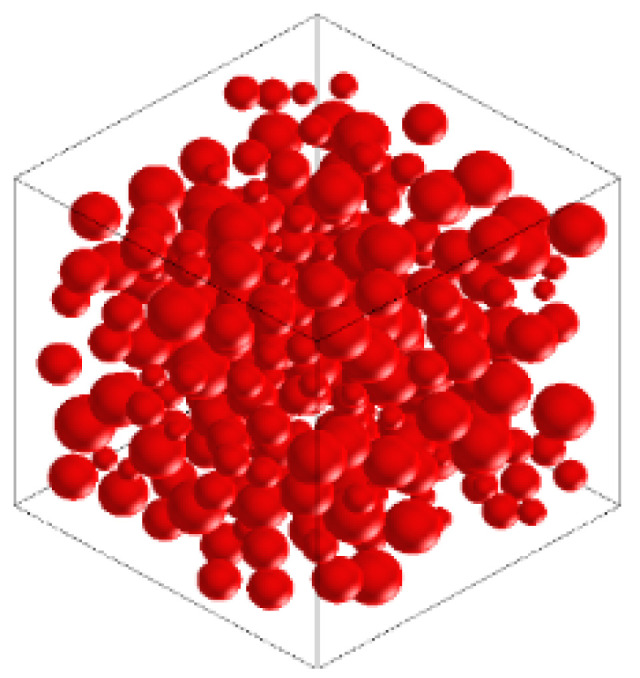
Three-dimensional random aggregate model of concrete.

**Figure 12 materials-15-03282-f012:**
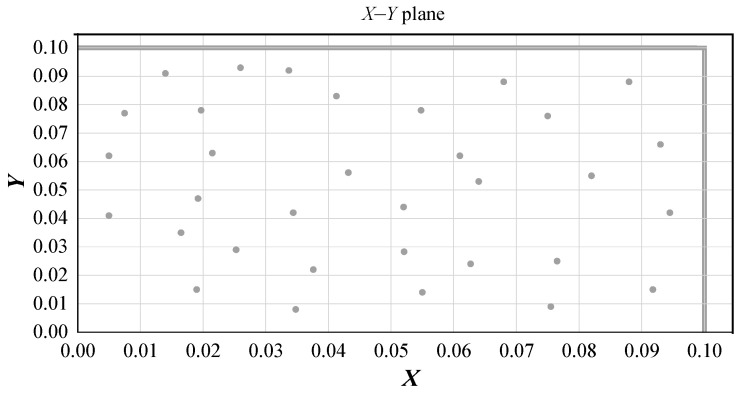
Distribution of aggregate in the X–Y plane in the 2D random aggregate model.

**Figure 13 materials-15-03282-f013:**
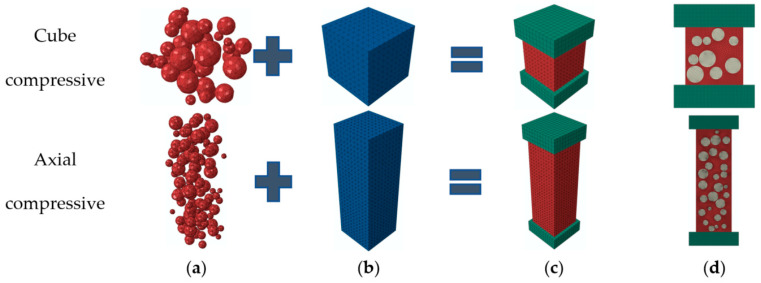
Random aggregate model of concrete: (**a**) aggregate, (**b**) mortar, (**c**) 3D model, and (**d**) 2D model.

**Figure 14 materials-15-03282-f014:**
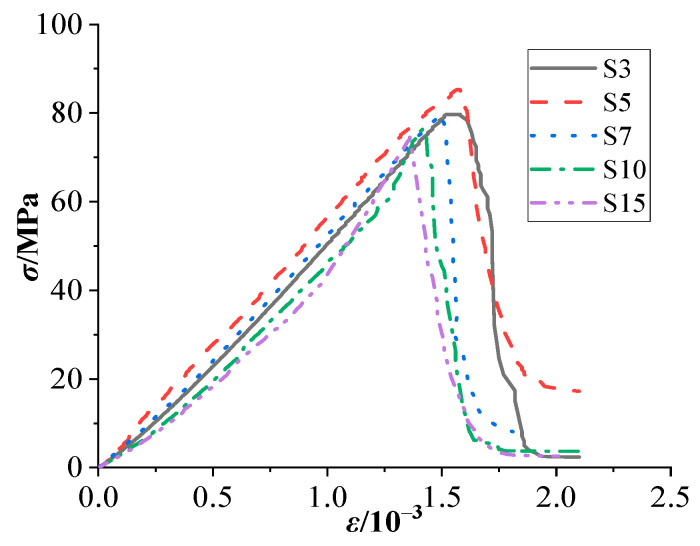
Stress–strain curves of concrete with different stone powder contents.

**Figure 15 materials-15-03282-f015:**
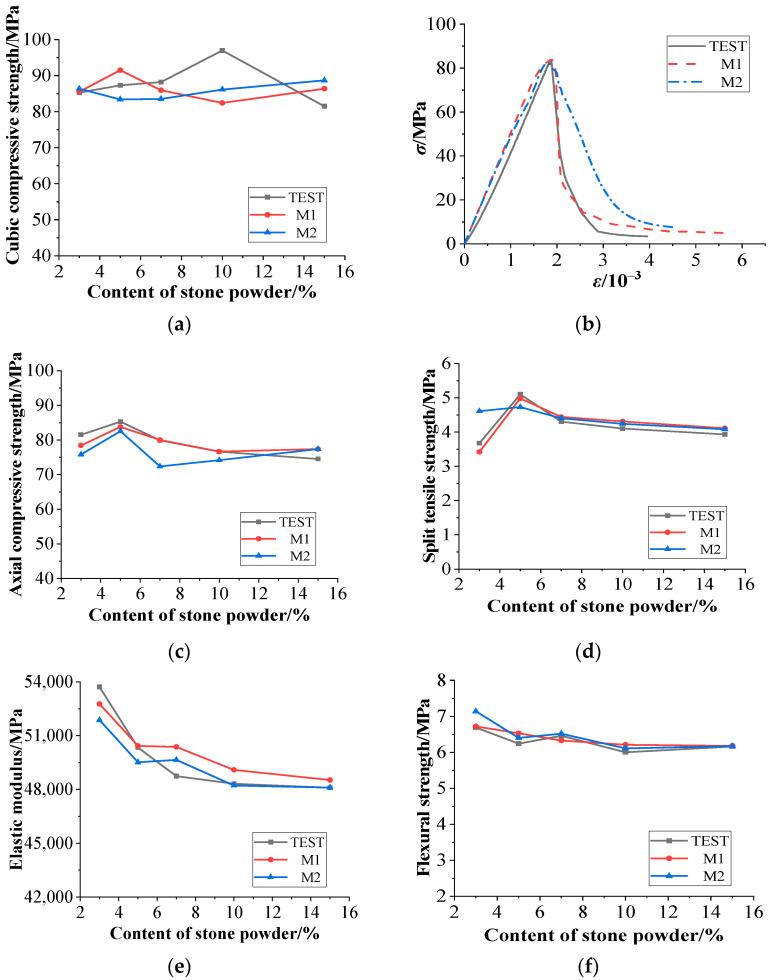
Comparison of experiment and simulation: (**a**) cubic compressive strength, (**b**) stress–strain curves of concrete with 5% stone powder content, (**c**) axial compressive strength, (**d**) split tensile strength, (**e**) elastic modulus, and (**f**) flexural strength.

**Figure 16 materials-15-03282-f016:**
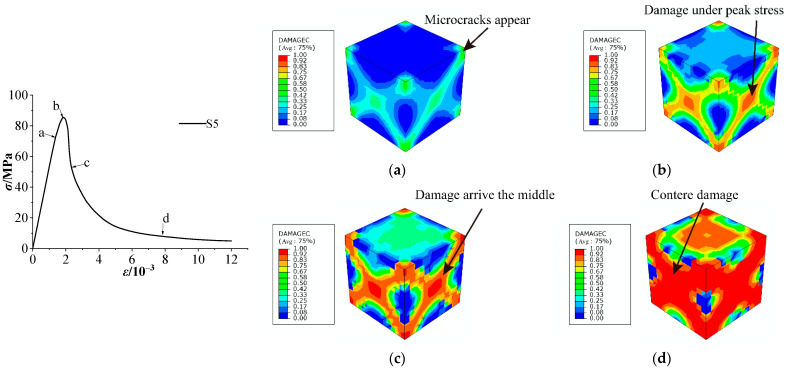
Cubic compression failure mode of concrete with 5% stone powder content (M1): (**a**) damage at point a, (**b**) damage at point b, (**c**) damage at point c, and (**d**) damage at point d.

**Figure 17 materials-15-03282-f017:**
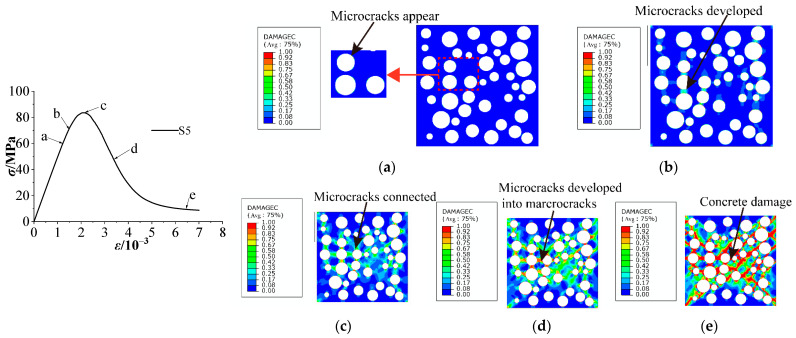
Cubic compression failure mode of concrete with 5% stone powder content (M2): (**a**) damage at point a, (**b**) damage at point b, (**c**) damage at point c, (**d**) damage at point d, and (**e**) damage at point e.

**Table 1 materials-15-03282-t001:** Main properties of cement and silica fume.

Cement	Specific Surface Area (m^2^/kg)	Volume Stability	IgnitionLoss (%)	Chloride Ion Content (%)	Initial Setting Time (min)	Final Setting Time (min)
P·O 42.5	321	qualified	3.95	0.014	170	219
Silica Fume	23,250	-	2.05	-	-	-

**Table 2 materials-15-03282-t002:** Performances of water-reducing admixture.

Test Items	pH	Chloride Content %	Total Alkalinity %
Test data	6.4	0.006	0.47

**Table 3 materials-15-03282-t003:** Performance test results of water reducing agent.

Test Items	Water-Reducing Rate (%)	Gas Content (%)	Bleeding Rate Ratio (%)	Compressive Strength Ratio (%)		Shrinkage Ratio %
				7 d	28 d	
Test data	28	2.8	30	155	147	99
Standard requirement	≥25	≤6.0	≤70	≥140	≥130	≤110

**Table 4 materials-15-03282-t004:** Chemical compositions of OPC, slag powder, fly ash, silica fume, and stone powder.

Chemical Compositions (%)	SiO_2_	Al_2_O_3_	Fe_2_O_3_	CaO	MgO	Na_2_O	TiO_2_	K_2_O	SO_3_
OPC	20.20	4.78	2.72	62.14	2.44	0.29	0.31	0.58	3.31
Slag Powder	34.50	17.70	1.03	34.00	6.01		-	-	1.64
Fly Ash	49.90	32.80	5.81	4.46	-	-	1.57	2.82	-
Silica Fume	96.10	0.39	0.18	0.19	0.12	0.09	-	-	-
Stone Powder	12.60	4.46	2.03	77.10	2.11	-	0.24	0.70	0.30

**Table 5 materials-15-03282-t005:** Mix ratio of C80 concrete (kg/m^3^).

Group	Rock Sand	Stone Powder	Coarse Aggregate	Cement	Water	Slag Powder	Fly Ash	Silica Fume
S0	709	0	1021	372	150	120	60	48
S3	687.73	21.27	1021	372	150	120	60	48
S5	673.55	35.45	1021	372	150	120	60	48
S7	659.37	49.63	1021	372	150	120	60	48
S10	638.1	70.9	1021	372	150	120	60	48
S15	602.65	106.35	1021	372	150	120	60	48

**Table 6 materials-15-03282-t006:** Multi-component 2D failure criteria.

Failure Mode	Failure Criteria
$\sigma_{11}\geq 0$	$F_{1}\left(\sigma \right)=\sigma_{11}/X_{t}$
$\sigma_{11}<0$	$F_{2}\left(\sigma \right)=-\sigma_{11}/X_{c}$
$\sigma_{22}\geq 0$	$F_{3}\left(\sigma \right)=\sigma_{22}/Y_{t}$
$\sigma_{22}<0$	$F_{4}\left(\sigma \right)=-\sigma_{22}/Y_{c}$
Shear failure	$F_{5}\left(\sigma \right)=\left|\sigma_{12}\right|/S$

**Table 7 materials-15-03282-t007:** Mechanical parameters of cement mortar (MPa).

Content of Stone Powder	Elastic Modulus of Mortar	Standard Deviation	Cv	Compressive Strength of Mortar	Standard Deviation	Cv	Tensile Strength of Mortar	Standard Deviation	Cv
S0	41,579	943.56	2.27%	72.4	2.84	3.92%	3.6	0.065	1.81%
S3	42,492	2866.25	6.75%	73.2	12.2	16.67%	3.8	0.35	9.21%
S5	40,757	1256.83	3.08%	70.2	4.5	6.41%	4.8	0.34	7.08%
S7	40,691	1691.23	4.16%	70.7	1.98	2.80%	4.1	0.18	4.39%
S10	39,934	5625.73	14.09%	73.6	4.54	6.17%	4.0	0.11	2.75%
S15	39,972	1731.58	4.33%	77.3	2.40	3.10%	3.8	0.12	3.16%

**Table 8 materials-15-03282-t008:** Cubic compressive, axial compressive, split tensile, and flexural strength of concrete with different stone powder contents (MPa) over 28 days.

Content of Stone Powder	Cubic Compressive Strength	Standard Deviation	Cv	Axial Compressive Strength	Standard Deviation	Cv	Split Tensile Strength	Standard Deviation	Cv	Flexural Strength	Standard Deviation	Cv
S0	83.70	2.05	2.45%	80.80	3.4	4.21%	3.65	0.21	5.75%	6.34	0.21	3.31%
S3	85.30	3.03	3.55%	81.50	5.65	6.93%	3.67	0.18	4.90%	6.69	1.27	18.98%
S5	87.30	9.9	11.34%	85.30	1.64	1.92%	5.10	0.37	7.25%	6.24	0.46	7.37%
S7	88.20	4.76	5.40%	79.90	7.87	9.85%	4.30	0.26	6.05%	6.46	0.19	2.94%
S10	97.00	2.79	2.88%	76.60	3.42	4.46%	4.10	0.22	5.37%	6.00	0.92	15.33%
S15	81.50	8.16	10.01%	74.50	3.80	5.10%	3.93	0.21	3.05%	6.16	0.14	2.27%

## Data Availability

Data sharing not applicable.

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
