# Peer review of "Experimental Mechanical Properties and Numerical Simulation of C80 Concrete with Different Contents of Stone Powder"

_materials, 2022, doi:10.3390/ma15093282_

Round 1

Reviewer 1 Report

The resubmission has addressed all issues noted in the first submission. 

The quality of the current manuscript is high and can be published as is.

Please add the units in y axis in figure 12c

Reviewer 2 Report

The paper reported the effect of stone powder content on the cement strength. The results were depended on experimental and theoretical approaches. The results outlined that the best percentage of stone powder in the C80 was 5% based on the strength data of various concrete shapes (i.e., cube, axial, and split). The manuscript was well written and contains a lot of information. So, I recommend accepting the manuscript after minor revision.

  • Figure 1b image is not clear, it is better to change it to be a clear image. In addition, in Figure A, it is highly recommended to add the distance label (i.e., 100 cm or m).
  • Please revise the manuscript language because there are some complex sentences.

Reviewer 3 Report

In general, the article can be reconsidered for publication after a major revision only. Overall, the paper is difficult to read; try to understand it using simple English.

  1. In the title of your paper, you state that you have focused on concrete, and in the abstract you have not used the word concrete, but instead you have used mortar. In my opinion, working on concrete specimens is much different from working on mortar. They are justly the same.
  2. What type of stone have you used? Perform a physical and chemical analysis of the stone powder.
  3. Include a particle size distribution curve for the stone that you have used in your work.
  4. Include the grade of concrete and code which you have referred to for conduction of testing of compressive strength, elasticity modulus, split tensile, and flexural strength of concrete in the 2.1 section.
  5. In section 2.1, water–cement ratio of 0.25, in my opinion, the water–cement ratio is too low, in spite of the fact that you are adding stone powder without admixture. In turn, the mix requires more water. Verify the same.
  6. Table 1, parameters which you have listed along with the results Include the code that you have followed to conduct the tests.
  7. In table 2, columns 3 and 4, stone powder I understood, but column 4 stone, what did you want to say?
  8. Figure 1b is not clear, include a better resolution figure.
  9. Table 3. How do you conclude the standard deviation and Cv are justified?
  10. Section 2.3.2 to 2.3.5, only experimental data you have written. Refer to the literature paper and include the analysis of the same. More explanation is required.
  11. Which software was used for the analysis in the Numerical Model Based on Inclusion Theory section?
  12. In section 3.1, Equations 4 and 5, how did you arrive at the equation? Write the process of the same.
  13. In section 3.1, alpha and beta, you have taken 1 and 10, justify.
  14. What is the significance of Matzenmiller–Lubliner–Taylor (MLT) in your research?
  15. What conclusion can be drawn from Figure 7? Include its significance and how silica fume affects the strength development of concrete through the model?
  16. It’s just software analysis, no experimental work you have carried out. What you have said in Section 3.2 “distribution of coarse aggregate had little effect on the mechanical properties of concrete”. Compare the experimental data (perform) with the software and come up with the percentage of error between them.
  17. Random Aggregate Distribution Model, you described about ITZ, then what about crack?
  18. The process and procedure for conducting analysis for the numerical models based on Inclusion Theory and Random Aggregate Distribution Model are included in Section 2. Materials and Methods.
  19. In Section 2, include the flowchart representing the methodology of your work.
  20. In section 4.1, you have explained about cracks with model analysis, but this analysis can be fully justified using only software, so I suggest you include SEM analysis in the ITZ phase showing cracks formation zone and discuss the same.
  21. In the discussion section, add previous literature to support your analysis.
  22. In the Discussion section, we highlight the performance of stone and silica powder content variations in compressive strength and axial compressive strength.
  23. In the entire paper, you have used the word "mortar" more than "concrete," so I suggest you include "mortar" in the title also.
  24. The physical properties of silica fume are included in table 1.
  25. Include more specific results in the abstract and conclusion.

Reviewer 4 Report

In this manuscript, the experimental mechanical properties and their corresponding numerical simulation of C80concrete were investigated. The manuscript is well written and is of great importance in the concrete field of research. Some questions and suggestions are provided below to enrich the scientific level of the presented manuscript before its publication:

  • It is suggested to change the title to “Experimental Mechanical Properties and …”.
  • In the abstract, the authors mentioned “The best stone powder content was 5% for C80 high-strength
    concrete by comprehensively considering its consistency”. Did the author mean the best dosage of stone in which the difference between experimental and numerical results are consistent or the best dosage for obtaining the larger amounts of strengths?
  • Are the artificial intelligence (AI) and machine learning (ML) techniques also capable of modeling the mechanical behaviour of the same concrete? If yes, they should be reviewed in the literature review.
  • The literature review could be enriched by adding the following references in the Introduction section:
    • Nafees, A., Amin, M. N., Khan, K., Nazir, K., Ali, M., Javed, M. F., ... & Vatin, N. I. (2021). Modeling of Mechanical Properties of Silica Fume-Based Green Concrete Using Machine Learning Techniques. Polymers, 14(1), 30.
    • Karimipour, A., Jahangir, H., & Eidgahee, D. R. (2021). A thorough study on the effect of red mud, granite, limestone and marble slurry powder on the strengths of steel fibres-reinforced self-consolidation concrete: Experimental and numerical prediction. Journal of Building Engineering, 44, 103398.
    • Golafshani, E. M., & Behnood, A. (2021). Predicting the mechanical properties of sustainable concrete containing waste foundry sand using multi-objective ANN approach. Construction and Building Materials, 291, 123314.
  • It is suggested to add figures of materials such as stone powder to the manuscript. XRF analyses results can be helpful.

Round 2

Reviewer 3 Report

The paper titled “Experimental Mechanical Properties and Numerical Simulation of C80 Concrete with Different Contents of Stone Powder”. After looking at the manuscript, I suggest that the author responds to these suggestions.

  1. Why is there such a variation in results for dosage of stone powder, compressive strength, axial compressive strength, and elastic modulus?
  2. Compressive strength, axial compressive strength, and elastic modulus mention the codes you referred to for conducting tests.
  3. Cite more recent papers. Consider and include all these references as they may benefit your article.
    1. doi.org/10.1016/j.matlet.2021.131302
    2. doi.org/10.1016/j.jmrt.2020.04.077
    3. doi.org/10.3390/ma14154304

Author Response

This manuscript is a resubmission of an earlier submission. The following is a list of the peer review reports and author responses from that submission.

Round 1

Reviewer 1 Report

The authors describe the influence of the stone powder content on the mechanical properties of concrete based on numerical simulation methods and experimental analysis. The results are interesting; however, some major issues should be addressed prior to the publication.

  1. The biggest drawback of the work is the lack of any statistical analysis of the obtained test results. The number of analyzed samples is unknown. The authors present only single results (perhaps average values) of the measurements of the mechanical parameters of cement mortars and concretes (Table 3, Table 4 and Table 5). Blank test (S0) should also be performed. In my opinion, the lack of an appropriate statistical test disqualifies this work (kind of very interesting) in its present form.

  2. Producers of cement and equipment used for samples testing were not mentioned.

  3. Axis titles on most charts are unreadable, I recommend increasing the font size.

Author Response

请参阅附件

Reviewer 2 Report

The originality and the scientific value of the subject research are good.
The research area is mechanical properties tests and numerical simulation for C80 concrete.

The manuscript has the usual structure, but part of the discussion is missing!!!

The chosen approach and procedure for numerical modelling is possible.
However, more information must be provided.

The overall similarity of the calculation and the informative value must be substantially improved. 
It is necessary to provide detailed information about the computational model,  parameters of the solver and boundary conditions. 
It would be appropriate to take more account of the stochastic nature of the material properties of concrete.

Provide a clear test scheme for mechanical properties in Tables 3 and 4.
In particular: Elastic modulus of interface Compressive
strength of interface; Tensile strength of interface

Statistical characteristics (standard deviation, VoC) are not clear.
Equations 1 to 3 are very basic. (may remain).

Improve the labels for the variables in Table 6 and then throughout the document.

Clearly state for which model the 2D or 3D calculation was performed?

What parameters influenced the creation of 2-D random models?
Were the properties the same across the thickness?

Make a clear reference to the model used - the constitutive model of concrete - DIGIMAT - Line 209

Figure 14 Improve image quality - some parts are difficult to see
Figure 14 complete description (a) ..... (e) ...

Overall, more calculations should be made to clarify the sensitivity (uncertainty) of the model and the effect of the input parameters for the concrete and its parts.

Were tests and modelling of structural elements done?

Overall, it is necessary to process the manuscript with greater interest.

Extensive research is underway in the area of nonlinear calculations of concrete structures when it is necessary to rework and expand the information in the introduction section. 
These are mainly the possibilities of material models of concrete, approaches to the choice of parameters, or taking into account the uncertainties in the calculation or stochastic character of concrete.
Sucharda, O. et.al. Non-linear analysis of an RC beam without shear reinforcement with a sensitivity study of the material properties of concrete. Slovak J. Civil Eng. 2020, 28, 33–43.
Valikhani, A.et. al. Numerical Modelling of Concrete-to-UHPC Bond Strength. Materials 2020, 13, 1379

The discussion chapter must be presented separately and present the results in the context of current research. What is the same, what is different?

A lot of references are in Chinese. Prefer English sources.
Reference 25 is red. Why?

Overall, it is necessary to improve the presentation of the results of numerical modelling and increase the informative value of the results.

The manuscript must be revised.

Author Response

请参阅附件。

Reviewer 3 Report

The paper presents the results fo experimental and numerical analysis of the influence of stone powder content on concrete characteristics. Despite the fact that paper is interesting, the reviewer has some remarks:

  1. The Introduction does not contain the clear aim of the paper. Also, there is no description of the original elements of the paper.
  2. The description of the experimental procedures is essential. It would be easier to follow for the reader if each section contains a short description of its content.
  3. The description of numerical simulation is too brief. Please add the comment about the material model, aggregate distribution (how the size of the aggregates and their position was determined - see the description of the model preparation in e.g. Numerical study of concrete mesostructure effect on lamb wave propagation).
  4. The article does not contain any statistical analysis - how many samples have been tested?

Author Response

请参阅附件

Reviewer 4 Report

The subject paper is interesting and its purpose complies with the journal’s aim and scope.

The authors are providing some interesting results on the addition of stone powder for the production of high strength concrete and also numerical modelling that can predict strength results with acceptable precision.

Some omissions have been identified and some necessary additions are suggested.

In terms of language usage, the manuscript is understandable, but needs proof reading by a native speaker.

More specifically:

Abstract: the authors should state what the stone powder is replacing in the mix.

Also what do you mean by liquidity? I believe the term you are looking for is “consistence”.

  1. Introduction:

It is important to state findings on other research on the addition of stone powder to high strength concrete mixes. Indicatively:

[1]      R. Yang, R. Yu, Z. Shui, X. Gao, J. Han, G. Lin, et al., Environmental and economical friendly ultra-high performance-concrete incorporating appropriate quarry-stone powders, J. Clean. Prod. 260 (2020) 121112. doi:https://doi.org/10.1016/j.jclepro.2020.121112.

[2]     H.F. Campos, N.S. Klein, J. Marques Filho, M. Bianchini, Low-cement high-strength concrete with partial replacement of Portland cement with stone powder and silica fume designed by particle packing optimization, J. Clean. Prod. 261 (2020) 121228. doi:https://doi.org/10.1016/j.jclepro.2020.121228.

[3]      Y.B. Yang, C.H. Man, W.S. Min, Y.X. Peng, C.K. Li, W. Bin Zheng, et al., Study on the C80 High-Strength Rock Chips Concrete, in: Ecol. Environ. Technol. Concr., Trans Tech Publications Ltd, 2011: pp. 218–223. doi:10.4028/www.scientific.net/KEM.477.218. 

One of the main issues with high strength concrete and mortar is consistence. You should discuss more on this.

  1. Materials and Methods

The mixes you have developed are not clear. What is your stone powder substituting?

How did you obtain the “optimal mix”? You are stating the “orthogonal method” but you give no explanations. Repeatability is important in all published research so you must provide details.

Table 3 and Table 4: how did you obtain these values? Shouldn’t they be in the “results” section of the manuscript?

Please provide a table for the S3, S5 etc mixes.

Test methods

The authors should explain the following:

  • which testing machines are used for each test,
  • at what rate the loading took place,
  • how many samples were tested per formulation,
  • if the mean value is provided and
  • what the standard deviation is

Line 259: “Then imported into the ABAQUS as a material parameter directly”: this is a very interesting statement. Please expand.

Lines 264-274: Damage index: please expand and describe what you are doing in section 2: Materials and methods

Author Response

请参阅附件

Round 2

Reviewer 1 Report

All comments have been addressed in the current version of the manuscript. 

Author Response

请参阅附件

Reviewer 2 Report

Thank you for the adjustments made.
The changes made the improvement of the manuscript.

The modifications are explained and commented on in detail. However, the final version of the document could be part of the submission.

It is necessary to revise and check the English language (eg.  MDPI).

The research area and results are from the context of the manuscript can better understand.

After the modifications, it is clear that the authors understand the issue and are well versed in it. The article will be interesting for the readers of the journal.

The manuscript contains all the main information.

The manuscript can be published in the journal.

Author Response

请参阅附件

Reviewer 3 Report

Thank you for addressing all reviewer's comments.

Author Response

请参阅附件

Reviewer 4 Report

The revised manuscript does not conform to the standards for revision:  A) it shows the raw form of review edits “on” with stricken through words etc super confusing and it seems that they managed to confuse themselves too, as they are deleting necessary information such as the standards of the tests they have carried out. B) the authors have failed to provide an analytical explanation of how and where (in which lines) they are addressing the reviewers’ comments. C) proof reading by a native speaker has not yet taken place

Author Response

请参阅附件
